# The Effect of Dialkyl Peroxide Crosslinking on the Properties of LLDPE and UHMWPE

**DOI:** 10.3390/polym13183062

**Published:** 2021-09-10

**Authors:** Pollyana S. M. Cardoso, Marcelo M. Ueki, Josiane D. V. Barbosa, Fabio C. Garcia Filho, Benjamin S. Lazarus, Joyce B. Azevedo

**Affiliations:** 1Department of Materials, University Center SENAI CIMATEC, Salvador 41650-010, BA, Brazil; pollyanam@fieb.org.br; 2Graduate Program in Materials Science and Engineering—P2CEM, Federal University of Sergipe (UFS), Aracaju 49100-000, SE, Brazil; mm_ueki@yahoo.com.br; 3Military Institute of Engineering, Rio de Janeiro 22290-270, RJ, Brazil; fabiogarciafilho@gmail.com; 4Materials Science and Engineering Program, University of Califórnia San Diego, San Diego, CA 92093, USA; bslazaru@eng.ucsd.edu; 5Institute of Science, Technology and Innovation, Federal University of Bahia, Salvador 42809-000, BA, Brazil; joyce.azevedo@ufba.br

**Keywords:** crosslinking, peroxide, LLDPE, UHMWPE

## Abstract

Peroxide has been considered a chemical agent that can be used to tune the properties of polymeric materials. This research evaluated the influence of different concentrations of dialkyl peroxides on the mechanical, thermal, and morphological properties of linear low-density polyethylene (LLDPE) and ultra-high molecular weight polyethylene (UHMWPE). The neat polymer, as well as those with the addition of 1% and 2% by mass of dialkyl peroxides, were subjected to compression molding and immersion in water for 1 h, under controlled temperatures of 90 °C. The values of the gel content found in the samples indicated that the addition of peroxide to the LLDPE and to the UHMWPE promoted the formation of a reticulated network. The structure obtained by the crosslinking led to less reorganization of the chains during the crystallization process, resulting in the formation of imperfect crystals and, consequently, in the reduction in melting temperatures, crystallization and enthalpy. The mechanical properties were altered with the presence of the crosslinker. The polymers presented had predominant characteristics of a ductile material, with the occurrence of crazing with an increased peroxide content.

## 1. Introduction

Crosslinking of polyethylene is considered one of the main approaches to enhance properties and meet application requirements that are not met by pure polyethylene [1,2].

Crosslinking occurs when the polymer adjacent chains become linked covalently. This bond can be formed directly through carbon–carbon bonds, or indirectly through a bridge-forming group, which creates crosslinks between the chains [3]. These crosslinks form from the combination of macroradicals generated by the abstraction of hydrogen from the polymer chain, resulting from the action of free radicals originating from the thermochemical decomposition of the product peroxides or from the irradiation of polyethylene [3,4].

Crosslinking can be initiated by physical and/or chemical agents. Chemical crosslinking is performed by adding an external agent such as azo, silane or peroxide. Several studies have reported the efficiency of peroxide as a crosslinking agent [5,6,7].

Peroxide promotes the thermal decomposition of polyethylene by releasing a radical that can remove hydrogen atoms from the polymer chain [8]. This type of reaction occurs rapidly until the peroxide is all consumed or until the temperature falls below the value required for the peroxide decomposition.

The crosslinking of polyethylene by organic peroxides has received attention in the literature and has become a widely accepted and studied method for modifying the microstructure and properties of polyethylene [9]. When this process uses peroxides in the modification of polyethylenes, crosslinks are formed by a reaction between macroradicals, which are produced by the thermal decomposition of the peroxide, followed by the removal of hydrogen from the main polyethylene chain [10].

Dialkyl peroxides (2,5-Dimethyl-2,5-Di(t-Butylperoxy)hexane) are considered as one of the most efficient chemical agents for the crosslinking of polyethylenes. The dialkyl peroxide decomposition mechanism is well established and involves breaking the O–O bond to form the alkoxyl radical, which is favored at higher temperatures [11].

In this work, crosslinking of LLDPE and UHMWPE, using the press-forming process, polymers with potential applications in the biomedical industry, was investigated. The process of crosslinking polyethylene using radiation is reported in the research as the most suitable for products intended for medical applications. In the present work, another crosslinking method for these products was evaluated. Chemically crosslinked polyethylene is well-known for applications where the product requires a high temperature resistance, such as wires and cables, or when the product is subjected to contact with hydrocarbon solvents and chemical products, where there is a tendency for the occurrence of cracks by tension. A crosslinked polyethylene (XPE) does not melt or flow above the melting point and has a better resistance to chemicals and hydrocarbon solvents [12,13].

Researchers have investigated the effect of reticulated LLDPE, using radiation, for applications in hip arthroplasty, instead of UHMWPE, due to its wear resistance [14,15,16]. UHMWPE is a polymer that combines specific properties, such as chemical inertness, biocompatibility, and good mechanical properties, which makes it suitable for a wide number of applications in biomechanics [17]. Although UHMWPE exhibits better wear resistance than many other polymers, the prolonged use in medical devices such as artificial junctions causes the shedding of polymer fragments in the human body. Several methods for increasing the wear resistance of UHMWPE have been studied. The crosslinking of UHMWPE with peroxide or radiation is an alternative, evaluated for modifying the wear of this material [18].

In this context, this research evaluated the influence of an alkyl peroxide concentration on the properties of different polyethylenes, LLDPE and UHWPE, during the crosslinking process with the goal of improving their functionality in the medical field. The mechanical properties under tensile and impact loading were analyzed. The gel content was determined in the compositions with 1% and 2% by mass crosslinking agent and the thermal properties were analyzed using differential scanning calorimetry (DSC). The morphology of the crosslinked polyethylene was evaluated after obtaining micrographs using scanning electron microscopy (SEM).

## 2. Materials and Methods

### 2.1. Materials

In this study, two types of polyethylene were used, linear low-density polyethylene (LLDPE) and high-molecular-weight polyethylene (UHMWPE). Both polymers are manufactured by Braskem (Camaçari, Brazil). The LLDPE was obtained from the commercial hexene comonomer ML 3601 and has a density of 0.939 g/cm^3^. The UHMWPE, commercially supplied as UTEC 6540, is supplied in a powder form, and has a density of 0.925 g/cm^3^.

The crosslinking agent was an alkyl peroxide, 2,5-Dimethyl-2,5-Di(t-Butylperoxy)hexane, produced by Akzo Nobel (São Paulo, Brazil), and is commercially known as Trigonox 101 XL in a powder form. The main properties of this crosslinker are: a density of 0.870 g/cm^3^; a molecular weight of 338; an active oxygen of 9.4; and a half-life of 10 h/119 °C, 1 h/138 °C, and 1 min/185 °C.

### 2.2. Methods

The polymers were manually mixed with the crosslinker at room temperature (RT). For this purpose, the LLDPE was micronized in a Tritumaq TM460 (Tritumaq, São Paulo, Brazil), at 1750 rpm, with sieves of 18 mesh and with the micronization camera temperature at 60 °C. The UHMWPE was used in the form supplied by the manufacturer. The particle size distribution of the polymers was determined by sieving according to ASTM D 6913-04. 

The concentrations of peroxide used were 1% and 2% by mass. The crosslinking process was carried out during the mold-pressing conformation. The conditions of mixing were defined after a dynamic oscillatory rheometry test. It was observed that no reduction in the viscosity was detected for 30 min at the temperature and times used. Therefore, it was decided not to use an antioxidant because the peroxide may have its efficiency reduced in the presence of this additive [19].

A hydraulic press, manufactured by Advance, was used under the following conditions: an electric resistance heated plateau; 60 kgf/cm^2^ of applied pressure; and an opening speed of 200 mm/s. A square mold of 195 mm × 195 mm × 4 mm was used to shape the polymers into a plate. To relieve the internal stress produced during the crosslinking process, the specimens were immersed in water for 1 h, with a controlled temperature of 90 °C.

The parameters used in the crosslinking process, time and temperature, were previously determined [6]. The LLDPE sample was subjected to crosslinking for 10, 20 and 30 min at temperatures of 140, 150 and 160 °C. Samples which were heated to 160 °C for 30 min showed values of gel content above 80%, which was found to be the ideal parameter for a high degree of crosslinking.

The mechanical characterization of the crosslinked polymers was carried out through tensile strength and impact resistance tests. In order to obtain consistent specimens of these tests, the temperature distribution was mapped in the hydraulic press used during conformation of the polymers. An FLIR T300 camera thermal imager (Instrutemp, São Paulo, Brazil) was used, with a temperature measurement in the range of −20 to 650 °C and with a thermal sensitivity of 0.05 °C. As for the mapping, the peroxide half-life temperature (138 °C) was considered for a time of 1 h. In this time, half of the amount of existing peroxide was decomposed. The most consistent temperatures were observed in the middle portion of the samples, which varied between 138 °C at the upper face and 140 °C at the lower face. Therefore, the specimens for analysis were removed from the central region of the mold; the temperature variation observed during the pressing did not influence the response of the analyzed properties. Figure 1 shows the temperature measurement points.

Tensile tests were performed on EMIC equipment model DL 200 (EMIC, São Paulo, Brazil), following ISO 527 type 5A, without the use of an extensometer and with a displacement rate of 50 mm/min. The resistance to the IZOD impact was determined on a pendulum impact machine CEAST INSTRON 9050 (INSTRON, Torino, Itália) according to ISO 180A, and a 2 mm deep notch at a 45° angle was made in each specimen. The hammer impacted each sample with 2.7 J of energy. In order to achieve a statistical validation, the results of the mechanical properties were obtained by the average of five samples for all the formulations studied.

The degree of crosslinking was determined according to the ASTM D 2765 standard. Initially, the material was weighed (W_inic_), and then extracted in xylene under reflux for 16 h at a temperature of 140 °C. Approximately 100 mL of xylene and approximately 0.25 g of the polymer were used in a 60-mesh screen. After extraction, the material was dried at 70 °C under a vacuum until it reached a constant weight (W_gel_). Using Equation (1), the gel content of the crosslinked material was calculated.
% gel = (W_gel_/W_inic_) × 100(1)

The thermal characterization of the crosslinked polymers was performed in a TA Instruments DSC model Q3 DSC (TA Instruments, New Castle, EUA) with a heating rate of 10 °C/min in a nitrogen atmosphere, in a temperature range from 23 to 200 °C, with two heating runs and one for cooling. Equation (2) was used to calculate the degree of crystallinity (X_c_), where the enthalpy of the 100% crystalline LLDPE used was 292 J/g and the enthalpy of the 100% crystalline UHMWPE was 293 J/g [3]. With the analysis, the melting temperature (T_m_), melting enthalpy (ΔH_m_) and crystallization temperature (T_c_) were determined.
(2)$$X_{c}=\frac{\Delta \mathrm{Hm}\;\left(\mathrm{amostra}\right)}{\Delta \mathrm{Hm}\;\left(\mathrm{PE}\;100\%\;\mathrm{cristalino}\right)}$$

The morphology of the polymers, before and after crosslinking, was investigated by scanning electron microscopy (SEM) using a JEOL microscope, model Carry Scopy JSM-6510LV (JEOL Ltd., Tokyo, Japan), with an acceleration voltage of 20 kV. The polymer powders and the fracture surface from the reticulated specimens were analyzed after impact resistance testing. The samples were mounted on an aluminum sample holder and covered with gold using Denton Vacuum evaporative metallization equipment, model DESK V (Denton Vacuum, Moorestown, NJ, USA).

## 3. Results and Discussion

Figure 2 shows the particle size distribution of polymers in powder form. It is observed that for LLDPE and UHMWPE, the particles have an average size distribution between 106 and 212 µm.

The SEM images show a clear difference in the shape and size of the particles (Figure 3). The LLDPE had a deformed particle appearance, which is different from the UHMWPE. The UHMWPE morphology consisted of particle aggregates with microcrazes that can be connected by fibrils. When the UHMWPE had been subjected to compaction and sintering, the literature indicates that the molecular free space is reduced and pores and weak connections form between the particles [20,21,22,23].

The typical stress versus strain curves for LLDPE and UHMWPE with the addition of 1% and 2% peroxide are shown in Figure 4. It was observed that for both polymers, the behavior is characteristic of a ductile material. For the LLDPE (Figure 4a), it was observed that with the addition of peroxide, there is an increase in deformation prior to rupture. This deformation causes an increase in stress corresponding to the increase in nominal stress as the molecular chains are stretched, characterizing the phenomenon called strain hardening [24]. For the UHMWPE (Figure 4b), it was possible to identify the strain hardening, where there was an increase in the resistance to deformation by plastic flow [21,24]. Strain hardening is strongly affected by the molecular weight [25]. 

In Figure 5, it can be observed that the breaking strength was affected by the tensile capacity of the polymer before breaking. In addition, a decrease in the breaking strength occurred for UHMWPE with the addition of 2% peroxide, around 36%, compared to pure UHMWPE, whereas for LLDPE there was an increase in this property. The polyethylene that suffered strain hardening during the deformation was shown to have a greater resistance to rupture than those that do not exhibit this behavior [26]. For the analyzed samples, at the beginning of the deformation, the LLDPE displayed a strain-softening behavior, and at the end, it transitioned to a strain hardening behavior, whereas the UHMWPE deformed less before breaking and presented a strain hardening behavior from the beginning of the flow, for this aspect that it presents. As a result, there was a decrease in the tension-tensile stress in the rupture with the increased reticulation. 

It was observed that the UHMWPE presented a decrease in the deformation with the addition of peroxide, whereas the LLDPE exhibited an inverse behavior (Figure 6). The restriction imposed on the elongation of the polymer increased with the increase in chemical crosslinks, which could be identified in the UHMWPE [27]. This restriction was due to the smaller length of the segments that provided deformation and the degree of crystallization, which also caused a decrease in the elastic modulus values [28,29,30], as shown in Figure 7.

The reduction in the elastic modulus with the addition of peroxide (Figure 7) is justified by the lower degree of crystallinity or the existence of crosslinks in the amorphous regions furthest from the crystallites [31]. Another aspect associated with the decrease in the Young’s modulus is the relationship between the modulus of the amorphous phase and the crystalline phase, where the modulus of the crystalline phase must be smaller than the modulus of the amorphous phase [26]. Previously, some authors reported a decrease in the mechanical properties for crosslinked polyethylenes (LLDPE and UHMWPE); however, an increase in wear resistance has been identified with the introduction of crosslinking between molecules [32,33,34]. The crosslinking allows the semicrystalline polymer to behave mechanically, similar to rubber above its melting temperature (T_m_), although still exhibiting properties of a thermoplastic below T_m_ [27].

The mechanism of the plastic deformation of semicrystalline polymers requires investigation due to the complex hierarchical architecture of such materials. The deformation of a semicrystalline polymer is a process that involves the crystalline lamellae, as well as amorphous layers without order. At temperatures where the amorphous phase exhibits rubber-like properties (T_d_ > T_g_), it is in the interlamellar regions that the initial deformation stage occurs. Therefore, the stress required to initiate the deformation of the amorphous phase constitutes from 2% to 10% of the stress necessary to activate the deformation mechanisms of the crystalline phase [33,35]. When the modification of the amorphous phase occurs, there is a direct relationship between the yield stress and ductility of the modified polymer. For LLDPE and UHMWPE chemically altered with peroxide, there is a direct change in the amorphous phase, causing a decrease in yield stress and an increase in deformation [35]. 

Figure 8 shows the results of the impact resistance tests. There was an increase in impact resistance with the addition of peroxide for both LLDPE and UHMWPE. This was due to the increase in molar mass and chain connectivity and the decrease in the degree of crystallinity, provided by the crosslinks, which promoted a resistance to crack propagation [5]. For the UHMWPE modified with peroxide, the impact resistance increased by approximately 62% relative to the pure UHMWPE. For the modified LLDPE, this improvement was more accentuated, with impact resistance increasing by around 300%. The more tenacity behavior occurred due to the change in the thickness of the lamellae, where the thinner lamellae resulted in smaller spherulites, causing a refinement in the morphology. The refined morphology induced an increase in impact strength and a decrease in crystallinity and lamella thickness. The flow stress and modulus should decrease, causing an increase in deformation at the break. However, the introduction of crosslinks in UHMWPE removed the deformation at the break, limiting the plastic deformation and being more striking; therefore, the molar mass was more significant [29,36].

The degree of crosslinking of the samples was determined by the gel content, and the results are shown in Figure 9. We observed that with the addition of peroxide in the polyethylenes (LLDPE and UHMWPE), the percentage of gel content showed a small increase. Polyethylene crosslinking involves the decomposition of peroxide when two segments of polymer molecules form a ‘cage’ around that molecule. The increase in the gel content achieved for UHMWPE in relation to LLDPE may be due to the higher entanglement density; it has a higher average molecular weight than LLDPE [37]. 

Some other thermomechanical factors can be considered for the degree of peroxide incorporation in polyethylenes. One is that during the mixing process, the peroxide and the polymer are in a powder form; therefore, the deposition of peroxide on the surface of the particles must be considered. When the mixture is heated, the peroxide decomposes on the surface instead of diffusing into the particles. As the peroxide content increases, the relative amount of decomposition on the surface of the particles increases as well. Heating during the pressing process compounds this trend. At this stage, peroxide can be decomposed on the surface of the particle, causing premature crosslinking and thereby decreasing compaction during processing [37]. This phenomenon may explain the small variation in the gel content with the increase in the percentage of peroxide (1% to 2%). It was also observed that for pure polyethylenes (0% peroxide), a gel content is identified that can be attributed to a certain degree of degradation from the extraction using the xylene solvent. The literature indicates that exposing the polyethylene to some extraction solvents for long a period of time can degrade the polymer [38].

The thermal properties of LLDPE and UHMWPE were determined with differential scanning calorimetry (DSC), where the melting enthalpy of the crystalline phase (ΔH_m_), melting temperature (T_m_) and the degree of crystallinity (X_c_) were measured.

Figure 10 and Figure 11 show the thermograms in the temperature range where the melting or crystallization events occurred for LLDPE and UHMWPE with the addition of 1% and 2% peroxide, respectively.

The melting enthalpy decreased with the addition of 1% peroxide for the two analyzed polyethylenes and slightly increased again with the addition of 2% peroxide on heating, as shown in Table 1. 

The decrease in the melting temperature was indicative of the formation of thinner crystalline lamellae, caused by the increase in the peroxide content [39,40,41,42,43]. In order to determine the thickness of the coverslip, Equations (3) and (4) were used.
(3)$$L_{c}=\frac{2\sigma_{e}T_{m}^{0}}{\left({\Delta H}_{m}^{0}\Delta T\right)}$$
(4)$$\Delta T=T_{m}^{0}-T_{m}$$
where L_c_ is the thickness of the lamella, σ is the free energy of the lamellar basal surface (for polyethylene it is 9 × 10^−6^ J/cm^2^) [39], T_m_ is the equilibrium melting temperature (for PE, 418 K) [39], ΔH_m_^0^ is the heat of fusion per unit volume (for PE, 280 J/cm^3^ ) [39], and T_m_ is the melting temperature obtained by the DSC.

Figure 12 shows the calculated lamella thickness results, where it was possible to identify the decrease in thickness with the addition of peroxide. In LLDPE, the lamella thickness decreased by 28% with the addition of 2% peroxide, whereas for the UHMWPE, the reduction in thickness occurred when 1% peroxide was added, with no significant change in thickness when this percentage was increased to 2%. The decrease in the thickness of the lamella can be associated with fragmentation of the lamellae caused by the crosslinking process [44]. The decrease in the lamellae thickness directly affected the mechanical properties, such as elastic modulus, tensile strength, ductility, and fatigue resistance, supporting the hypothesis that the mechanical properties depend on crystallinity [33,40].

Figure 13 shows the influence of the peroxide content on the crystallinity of the LLDPE and UHMWPE samples. The degree of crystallinity decreased with an increased crosslinking relative to neat polyethylene; however, much smaller differences were observed between the 1% peroxide and 2% peroxide samples. This variation in the degree of crystallinity occurred due to the formation of cross bonds while the polymer was in the amorphous state. Once formed, these bonds make it difficult to reorganize the polymer chains during the crystallization process, which results in the formation of imperfect crystallites with a smaller size [26,45,46,47].

The degree of crystallinity value for neat LLDPE was identified at around 52%, which is in line with that found by other authors [26]. The reticulated network formation in polyethylene causes difficulties in the reorganization of the chains during the crystallization process. This leads to the formation of imperfect crystals and, consequently, a reduction in the melting and crystallization temperatures, enthalpy of melting, and crystallinity. This can also lead to changes in properties that directly depend on the crystallinity of the material, such as tensile strength and modulus of elasticity [28,47].

Figure 14 shows micrographs of LLDPE fracture surfaces of samples treated with 0%, 1% and 2% peroxide, respectively. For samples crosslinked with 1% peroxide, it was possible to identify some areas with characteristics of a brittle fracture. For a lower portion of the region with respect to a more ductile portion, which is in accordance with the content of gel (82% in gel) measured, with a greater gel strength the morphology of the fracture approached the point fragility.

Commonly, polyethylene may exhibit two basic deformation mechanisms: multiple cracks (crazing) and flow shear (shear yielding). The brittle behavior results from the joining of multiple fissures and can be identified by its off-white shape [48]. The ductile behavior involves shear yielding. This phenomenon was identified in samples with the highest gel content (2% LLDPE peroxide), and characterized the transition from ductile to brittle behaviors as the crosslinking increased. For samples reticulated with 2% peroxide, ductile failure was much more prevalent with brittle failure only being noted in a few places. 

The images obtained of the UHMWPE samples are observed in Figure 15. The neat condition (0% peroxide) exhibited a brittle morphology, characterized by a package of microcable strips [48]. With the addition of peroxide, the predominant ductile material morphology was verified along with some brittle regions [49,50].

## 4. Conclusions

The addition of alkyl peroxide in LLDPE and UHMWPE altered the structure of the polymers, promoting the formation of a reticulated network, which changed the crystallization process, resulting in the formation of imperfect crystals and, consequently, in the reduction in melting temperatures, crystallization, and in the enthalpy of crystalline fusion. These changes also influenced the mechanical properties of the polymers, resulting in polymers with a lower elastic modulus and a greater resistance to impact.

The morphology of the LLDPE and UHMWPE with peroxide showed predominantly ductile characteristics with increased crazing with the increase in the peroxide content. This phenomenon could characterize a shift in the behavior of the material from ductile to brittle.

## Figures and Tables

**Figure 1 polymers-13-03062-f001:**
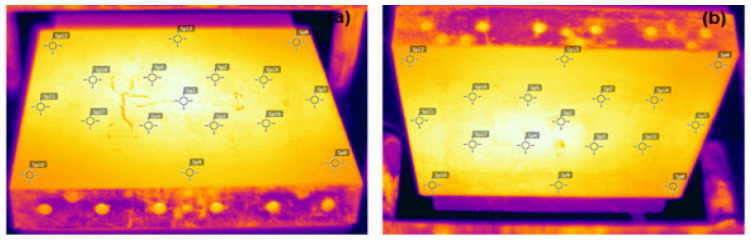
Thermal map of the plateaus of the press. (**a**) Lower and (**b**) upper.

**Figure 2 polymers-13-03062-f002:**
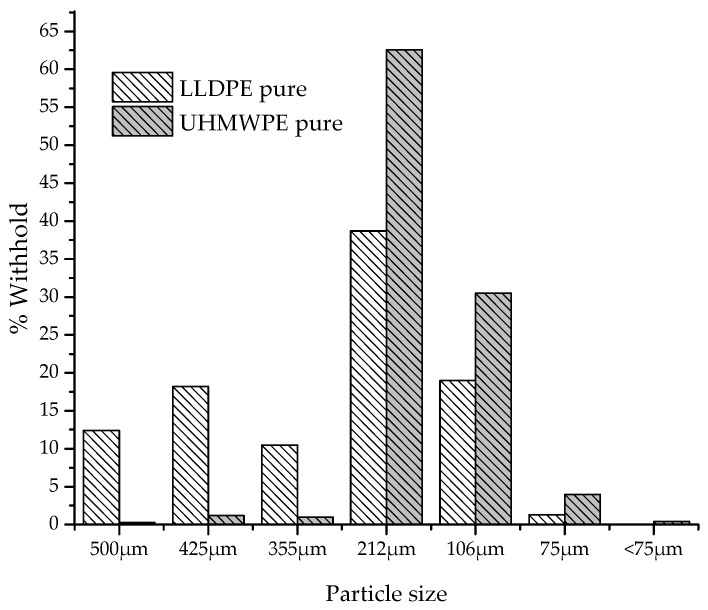
Particle size distribution of LLDPE and UHMWPE.

**Figure 3 polymers-13-03062-f003:**
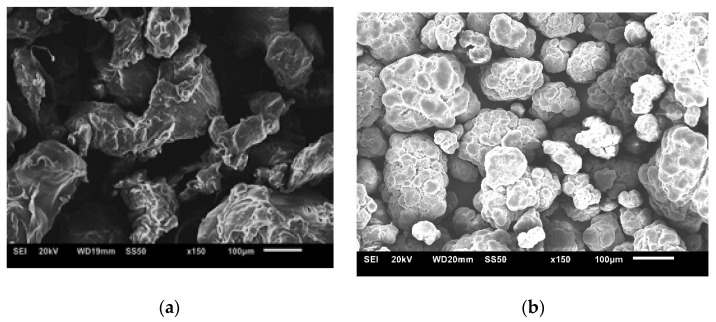
SEM images of pure polymer powders: (**a**) LLDPE; (**b**) UHMWPE.

**Figure 4 polymers-13-03062-f004:**
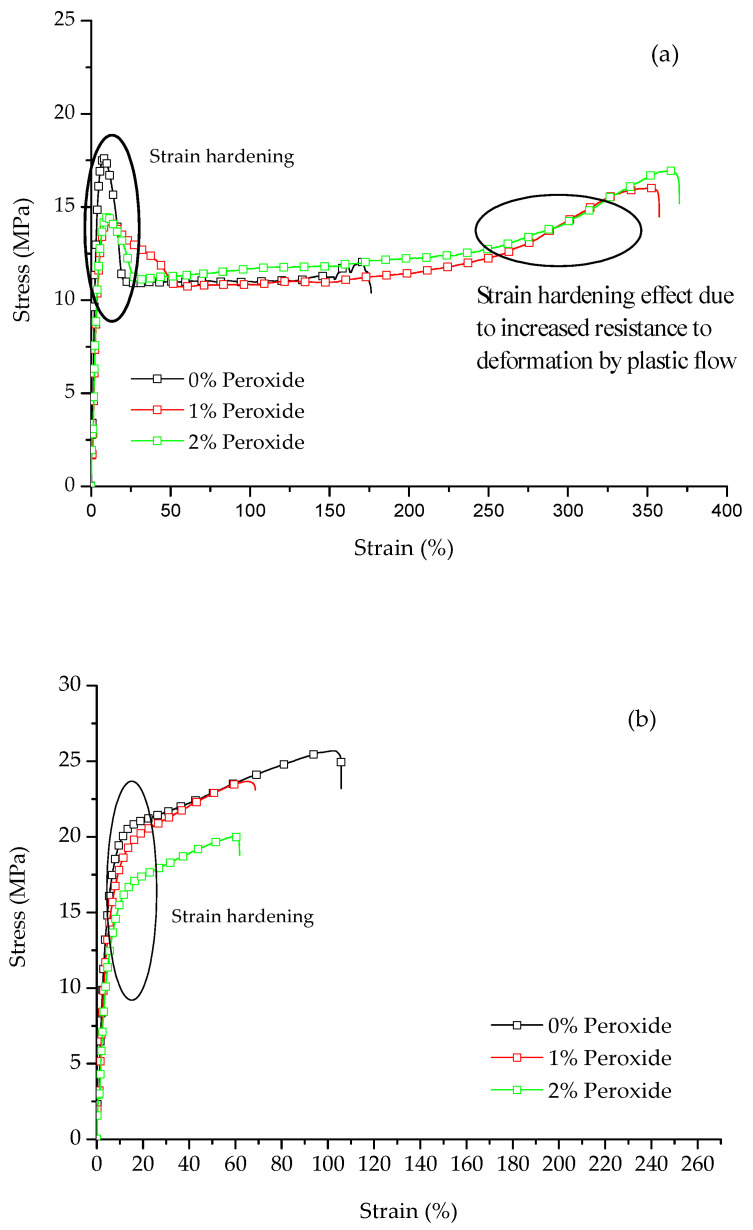
Typical strain versus stress curves for (**a**) LLDPE and (**b**) UHMWPE with the addition of 1% and 2% by mass peroxide.

**Figure 5 polymers-13-03062-f005:**
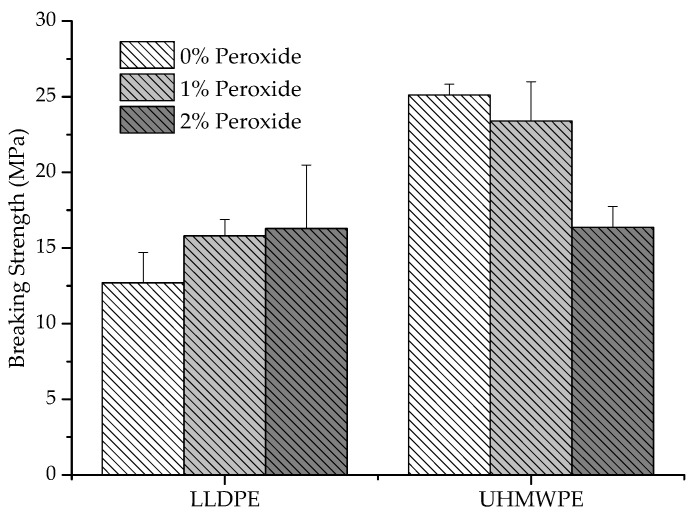
Variation of tension at the rupture of LLDPE and UHMWPE with 0%, 1% and 2% peroxide.

**Figure 6 polymers-13-03062-f006:**
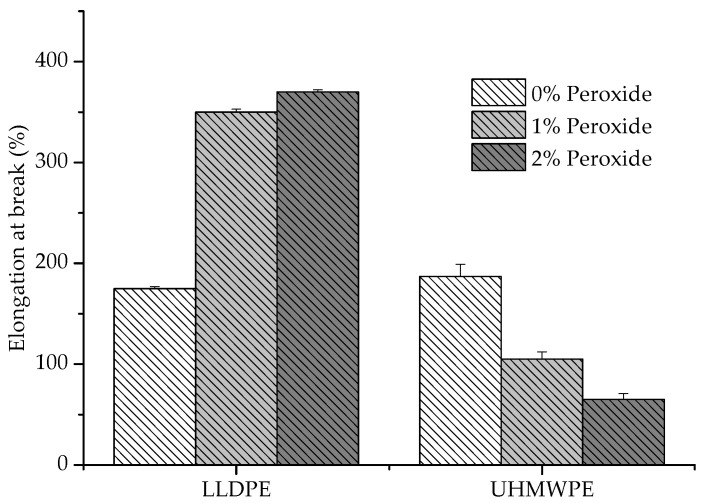
Deformation at the rupture of LLDPE and UHMWPE with 1% and 2% peroxide.

**Figure 7 polymers-13-03062-f007:**
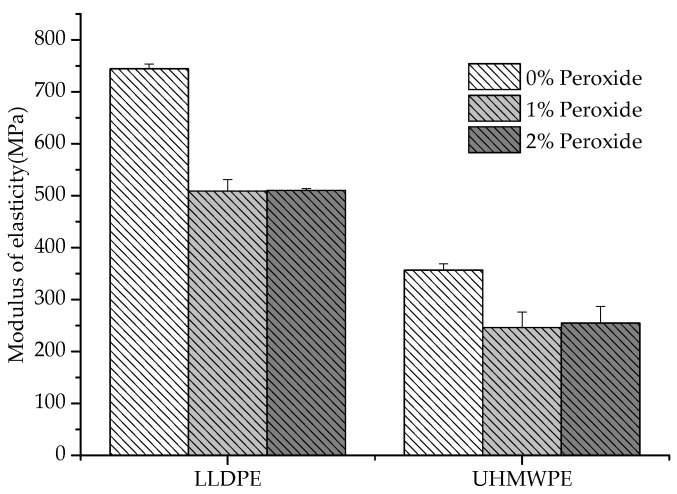
LLDPE and UHMWPE elastic modulus with 1% and 2% peroxide.

**Figure 8 polymers-13-03062-f008:**
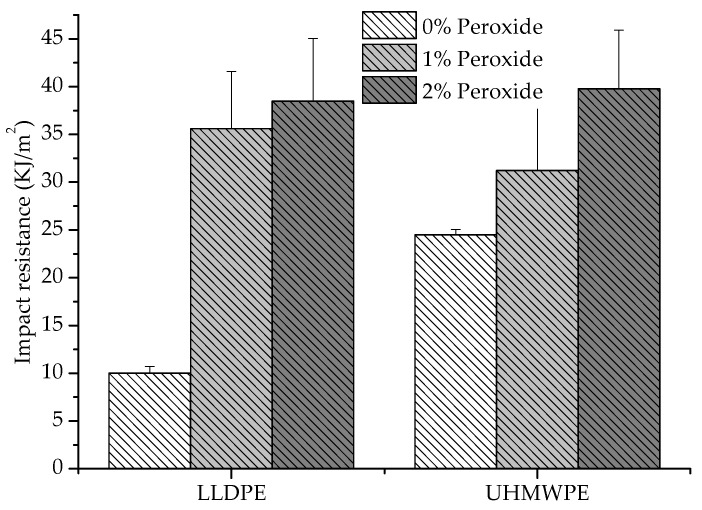
Resistance to the impact of LLDPE and UHMWPE with 1% and 2% peroxide.

**Figure 9 polymers-13-03062-f009:**
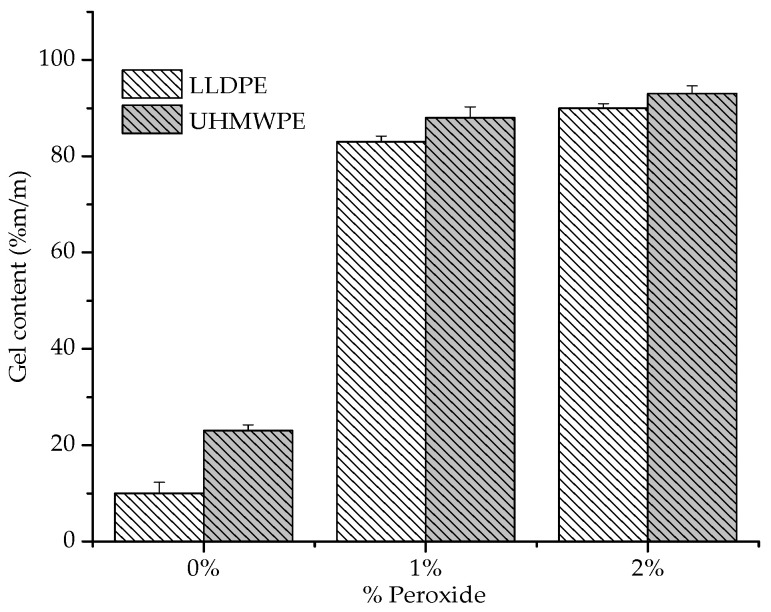
Gel contents of LLDPE and UHMW PE with 1% and 2% peroxide.

**Figure 10 polymers-13-03062-f010:**
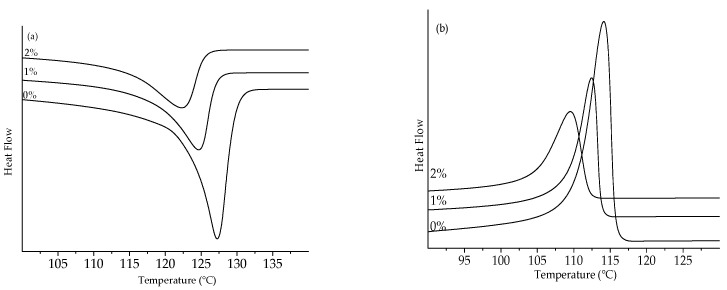
Thermograms of LLDPE peroxide: (**a**) melting peak and (**b**) crystallization peak.

**Figure 11 polymers-13-03062-f011:**
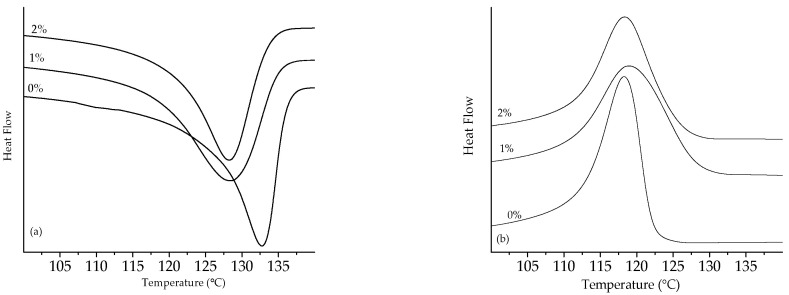
Thermograms of UHMWPE peroxide: (**a**) melting peak and (**b**) crystallization peak.

**Figure 12 polymers-13-03062-f012:**
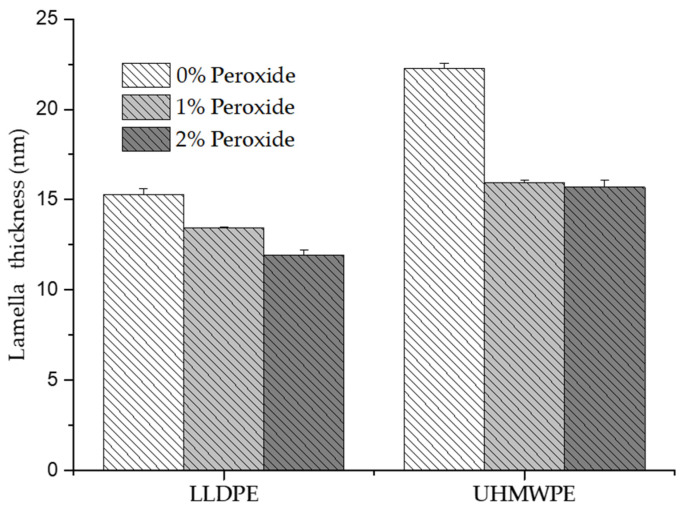
Thickness of the lamella determined by DSC for the LLDPE is UHMWPE with 0%, 1% and 2% peroxide.

**Figure 13 polymers-13-03062-f013:**
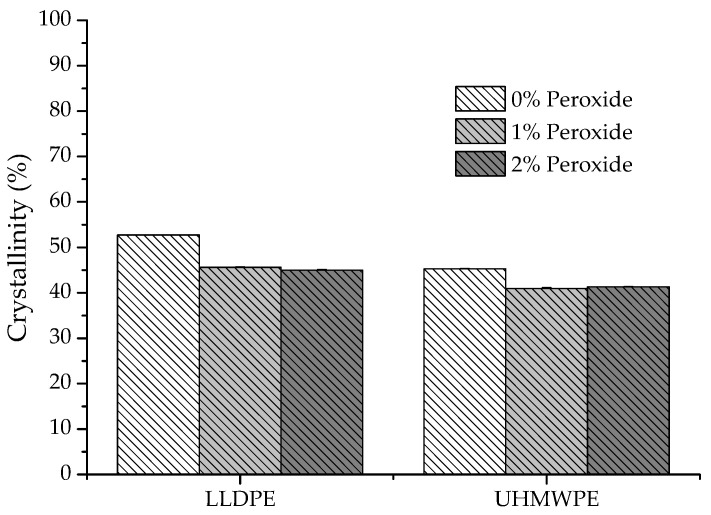
Variation in the degree of crystallinity of LLDPE and UHMWPE with 0%, 1% and 2% peroxide.

**Figure 14 polymers-13-03062-f014:**
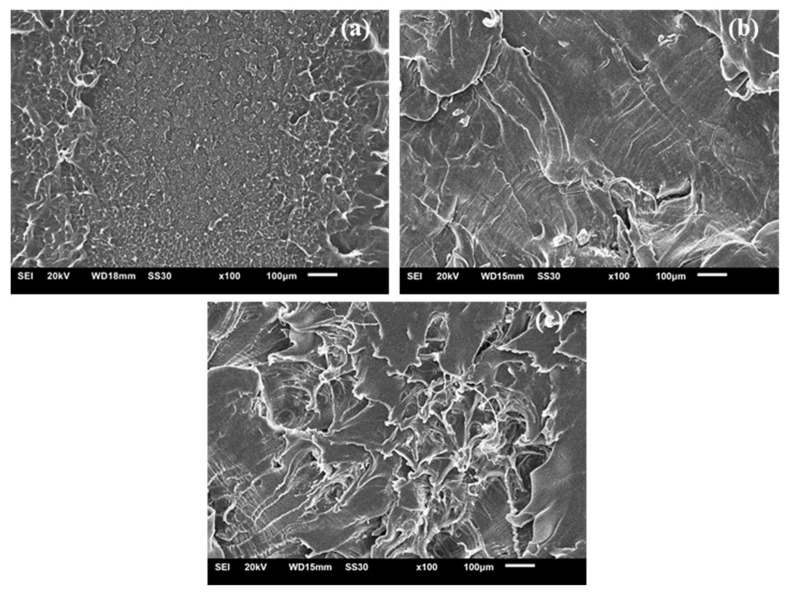
SEM images of crosslinked LLDPE (**a**) 0% peroxide, (**b**) 1% peroxide and (**c**) 2% peroxide.

**Figure 15 polymers-13-03062-f015:**
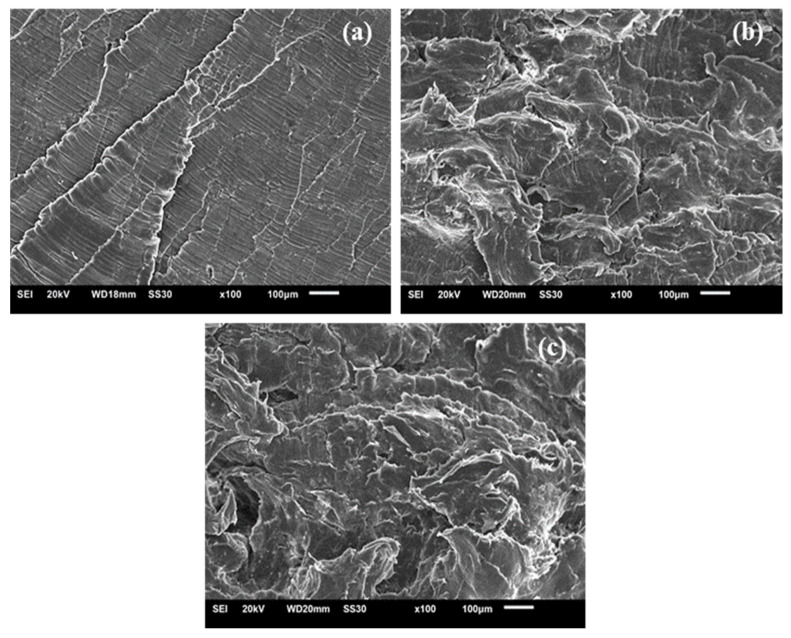
SEM images of UHMWPE with (**a**) 0% peroxide, (**b**) 1% peroxide and (**c**) 2% peroxide.

**Table 1 polymers-13-03062-t001:** Thermal properties of LLDPE and UHMWPE with 0%, 1% and 2% peroxide.

Condition	T_m_ (°C)	ΔH_f_ (J/g)	T_c_ (°C)	X_c_ (%)
LLDPE 0%	127.0 ± 0.4	153.1 ± 0.4	114.1 ± 0.1	52.7 ± 0.1
LLDPE 1%	124.8 ± 0.2	131.0 ± 0.4	111.6 ± 0.2	45.6 ± 0.2
LLDPE 2%	122.0 ± 0.1	138.7 ± 0.2	109.5 ± 0.2	47.0 ± 0.2
UHMWPE 0%	132.7 ± 0.9	125.0 ± 0.4	118.3 ± 0.1	45.3 ± 0.1
UHMWPE 1%	128.4 ± 0.5	117.8 ± 0.8	118.9 ± 0.2	41.0 ± 0.2
UHMWPE 2%	128.2 ± 0.7	123.3 ± 0.3	118.3 ± 0.2	42.1 ± 0.1

## Data Availability

Not applicable.

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
