# Peer review of "The Effect of Dialkyl Peroxide Crosslinking on the Properties of LLDPE and UHMWPE"

_polymers, 2021, doi:10.3390/polym13183062_

Round 1

Reviewer 1 Report

The authors present an interesting work analysing how the addition of peroxide modify the behaviour of two different types of PE. Overall, it is a simple interesting document, although it requires being more deep in different parts, specially in those related with the mechanical behaviour.

Specific comments:

Last sentence of the abstract is not correct.

I would provide references for the different standards used in the research.

The authors refers to “stretching”, but the concept is not clear

Figure 3: it would help to have both images with the same scale.

Figure 4: it is strain hardening, not softening. Besides, the different behaviour of the two materials should be further analysed.

Table 1: identify and define the different parameters.

Major issue: UHMWPE increases the R.I when adding peroxide, whereas its corresponding stress-strain curve is clearly more brittle. Authors should justify this, because it is contradictory.

Author Response

I am putting attached the response for the 3 reviewers.

Thanks 

Reviewer 2 Report

The reviewed work entitled: “Effect of Dialkyl Peroxide Cross-Linking on the Properties of LLDPE and UHMWPE” by Pollyana S.M. Cardoso, Marcelo M. Ueki, Josiane D.V. Barbosa, Fabio C. Garcia Filho, Benjamin Lazarus and Joyce B. Azevedo presents the results of the work of the modification of two types of polyethylene by dialkyl peroxide. In order to verify the results of the modification, the authors performed tests of mechanical properties, determined the degree of cross-linking and performed crystallinity tests by using the DSC method.
The discussed topic in the reviewed paper is known and widely used, which was also stated by the authors. Nevertheless, there is no emphasis on the new contribution to the development of the issue. The authors suggest medical applications of this type of materials, however, the research methods used are not corresponding with the designated application. 
Fig. 5 -fig. 8 are a repetition of table 1. Please choose one way of presenting the results.
 What is the different nature of maintaining the mechanical properties of the cross-linked LLDPE and UHMWPE?
 Does the mixture preparation process proposed by the authors not require the use of an antioxidant?

Author Response

(The authors gave the same response as above.)

Reviewer 3 Report

The manuscript by Cardoso et al. describes the use of a peroxide to induce crosslinking in linear low density- and ultrahigh molecular weight polyethylenes (LLDPE and UHMWPE), thereby affecting the crystallinity and mechanical properties.

I found no major problems with the manuscript, although there were some minor issues, including some instances where the text did not appear to match the data shown.

L94 and Fig. 1 caption: I think 'plateau' should be 'platten'.

L166-167:  The data in Fig 4b (UHMWPE) appears to only show strain softening.  Please check.

L198-199:  I think the modulus of the crystalline material should be higher than amorphous material.  Please check.

Subject to addressing those issues, I would be happy to recommend publication.

Author Response

I am putting attached the response for the 3 reviewers.

Thanks,

Round 2

Reviewer 1 Report

It can now be published. Typo in fig15 caption.

Reviewer 2 Report

I would like to thank the authors for clarifying my doubts. In my opinion, the changes made to the manuscript increase the scientific value of the work, and the re-reviewed manuscript in its current form can be recommended for further publication.